# Mediterranean Coastal Lagoons Review: Sites to Visit before Disappearance

**Juan Soria** [1,*] , **Rebeca Pérez** [1] **and Xavier Sòria-Pepinyà** [2,*]

1   Cavanilles Institute of Biodiversity & Evolutionary Ecology, University of Valencia, 46980 Paterna, Spain;
    repegon@alumni.uv.es
2   Image Processing Laboratory, University of Valencia, 46980 Paterna, Spain
*   Correspondence: juan.soria@uv.es (J.S.); soperja@uv.es (X.S.-P.)

**Abstract:** Coastal lagoons are an established priority habitat in the European environment because of the biological communities that inhabit them. Their origin is related to the transport of sediments from a nearby river or the movement of sands by the marine currents that produce the closure of a gulf. Therefore, they are recent geological formations, which also disappear quickly if environmental conditions change. The 37 coastal lagoons with a surface area greater than 10 km$^2$ located in the Mediterranean basin have been identified. Fishing has been the traditional use of these lagoons, in addition to their use as a navigation harbor when they are open to the sea. Pollution, quality problems and their consequences are the most studied topics in recent publications. Sentinel-2 images taken in the summer of 2020 have been used to study water transparency, suspended matter and chlorophyll *a* concentration. The result was that only six of them are in good ecological condition, but most of them are eutrophic due to the impacts on their environment and the inflow of poor quality water. The cultural values of these lagoons must also be protected and preserved.

**Keywords:** fishing; pollution; transparency; eutrophication; Mediterranean basin

## 1. Introduction

Coastal lagoons are shallow bodies of water, close to the sea, generally separated from it by a sandy bar that has produced the closure of an ancient marine gulf. They are mostly the result of the accumulation of sands and gravels of continental origin that are dragged to the coast by a river and whose accumulation is also due to the action of persistent currents. A continuous input of materials is necessary for the accumulation to occur, as well as the beach current favoring deposition in shallow areas, gulfs and inlets where there is also no major river current [1]. The balance between the contributions from its basin and its vicinity, the strength of the contributions from the interior and the accumulation by the sea are the three factors necessary for a lagoon's formation (Figure 1). Their presence is abundant on the coasts of the Mediterranean Sea, in the Baltic Sea and in some areas of the Atlantic (France, Portugal, Morocco, Brazil, Uruguay).

The Mediterranean Sea has its origin in the Tethys Sea, but in a totally different way from the one we know today, with most of the land surface we know today being flooded. About seven million years ago, the movement of the tectonic plates closed the Rifian and Betic corridors (previous to the actual Gibraltar strait), so that the transfer of water was severely affected and what is known as the Miocene saline crisis occurred, with the closure of the Mediterranean Sea [2]. The sea was left as an inland sea isolated from the Atlantic and fed only by the fluvial waters of its rivers. This resulted in a drop in sea level of hundreds of meters due to evaporation and direct communication between Africa and Europe through the southern Iberian and Italian peninsula. About five million years ago, the communication with the Atlantic Ocean was opened again through the current Gibraltar corridor, and once the same water level was recovered in the two seas, the formation of the coastline began, following the approximately present line.

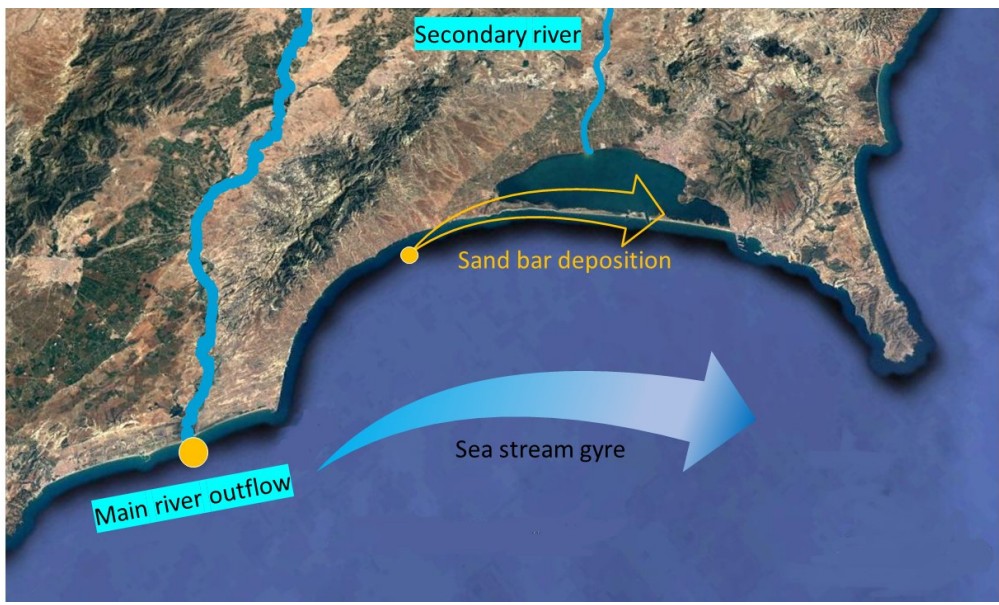

**Figure 1.** Ideal coastal lagoon formation scheme. Base figure: Google Earth.

Therefore, the geological origin of the current coastal lagoons is recent, a few thousand years old for the most part, and they are the result of coastal dynamics. Moreover, as with many lakes, they are ephemeral formations as they are created in a few thousand years and also disappear either by siltation of the sediments that reach them or by disappearance of the sandy bar by marine erosion [3]. All the Mediterranean coastal lagoons are from the most recent Holocene period, and other sub-current lagoons are known, such as the Elx lagoon in Spain, located about 15 km from the coast, which is a remnant of the ancient Mediterranean coast prior to the Miocene saline crisis [4].

The Mediterranean coastal zones exemplify the conflict between the human exploitation of water resources and the ecological needs of aquatic ecosystems [5]. The close relationship of lagoons with terrestrial ecosystem boundaries makes these environments very vulnerable to hydrological modifications (freshwater diversions or drainage discharges), water pollution and habitat loss [6–9], which deeply change the structure of the lagoons' ecological dynamics [10].

Coastal lagoons are a habitat declared as a priority by the European Union in the Habitats Directive, i.e., they are threatened with extinction [11]. This directive includes as coastal lagoons, in addition to those described as a typical lagoon formation, coastal evaporation salt marshes, which are also frequent on the shores of the Mediterranean. However, a hydrological distinction must be made between a coastal lagoon open to the sea and an estuary, considered as a tidal floodplain. This means that, for the purposes of the Directive, the classification of similar formations as coastal lagoons or estuaries will depend on the strength of the fluvial inputs that flow into the body of water and the influence of the tide on water renewal. This detail means that lagoons are very common in the Mediterranean due to weak tides and rivers with few contributions, while in the Atlantic the tides are higher and the rivers tend to have more contributions [12].

Therefore, coastal lagoons are bodies of water with scarce renewal due to low freshwater inputs and reduced communication with the sea. Their salinity is usually lower than the main sea when freshwater inputs are high and their volume is low; but this can change abruptly due to the influence of marine storms that overtake the dune bar and flood a lagoon with seawater. However, they can also be hypersaline when there is no freshwater input and the entry of seawater increases salinity by evaporation. In this sense, it can be said that they will have a positive estuarine circulation in the first case (fresh water flows out) or a negative estuary (marine water enters); but let us not forget that they are not estuaries in any case (Figure 2) [13].

The present work provides a compilation of Mediterranean coastal lagoons, with their essential morphometric data and their trophic state evaluated by satellite images during the summer of 2020, employing automatic procedures using satellite images contrasted and validated in numerous works. All this with the aim of obtaining a representation of the state of a threatened habitat at the European level.

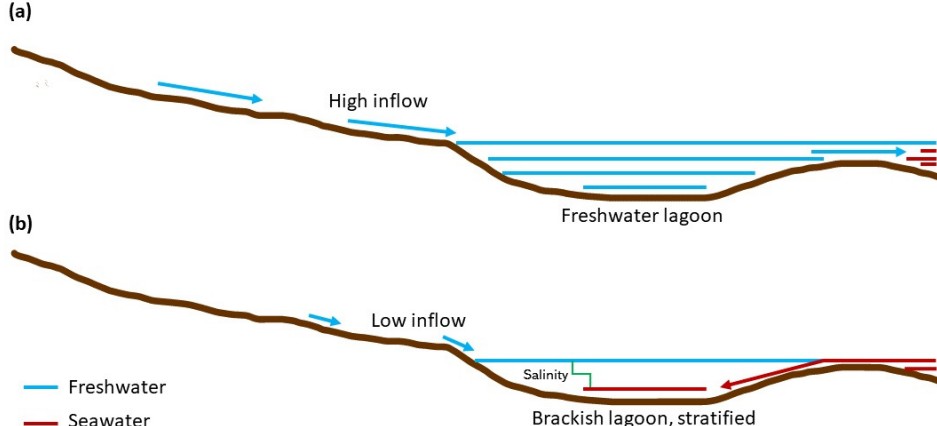

**Figure 2.** Water circulation models: (**a**) in high freshwater inflow; (**b**) in low inflow, with seawater entrance and high salinity lagoons.

## 2. Materials and Methods

The main coastal lagoons considered in this study are those whose surface area is larger than 10 km$^2$. Starting from the city of Marseille (France), and going counterclockwise along the coast of the Mediterranean Sea and its inland seas, the lagoon formations that have more or less close contact with the sea have been reviewed. Using the Google Earth Engine application (https://explorer.earthengine.google.com/, accessed on 15 January 2022), the Landsat-8 image was obtained and the Sentinel-2 image was downloaded for each lagoon during the summer of 2020 in an image free of clouds as much as possible (if there were none, it was searched for in spring 2020) and the reflectivity bands were obtained from it.

Using the Google Earth tool, the coordinates of the lagoon's barycenter were located. The polygon was drawn and the perimeter of the external area of the wetland was obtained, including the border vegetation, and the closed surface that includes the interior islands if they exist but excludes anthropized and urbanized areas. Likewise, the widths of the communication mouths with the sea were measured, adding them together in cases where there are several mouths (the most common). The maximum depth was obtained from Google Earth measurements, and they were reviewed with accredited bibliographic references [14–23].

Using the Scopus bibliographic tool, a bibliographic search was carried out using "lagoon" and the name of each site as keywords, from 1970 to 2021, and then selecting the most significant references for each site according to the Scopus classification.

Sentinel-2 images of each lagoon for spring or summer 2020 from the Copernicus Hub repository were used to obtain the automatic values using the SNAP application (Brockmann Consult Gmbh, Hamburg, Germany). These images were downloaded in L1C format, without atmospheric correction. The atmospheric correction was processed using the Case 2 Regional Coastal Color tool together with the Case 2 Extreme neural networks for complex waters, after resampling the images so that their bands have the same spatial resolution of 10 m. Once the resampling and correction procedure was completed, the automatic SNAP products were obtained, which are the concentration of Chlorophyll *a* (CHL), the concentration of total suspended matter (TSM) and the transparency of the water measured as the coefficient kd_z90max, the depth at which 10% of the incident

radiation reaches the surface. The values obtained with this procedure have been validated by numerous published papers.

Statistical treatment of the results was performed using Excel spreadsheet (Microsoft Corporation, Redmond, CA, USA).

## 3. Results

The geographic search provided a list of thirty-seven water bodies of the minimum dimensions considered as coastal lagoons, which are listed in Table 1. Twenty-two lagoons are located on the European continent, eight on the African continent and four are in Asia, specifically in the Sinai and Anatolian Peninsulas. With regards to marine sub-basins, eight are located in the Lion Gulf, two in the Balearic Sea, three in the Alboran Sea, three in the Sicily Sea, two in the Gabès Gulf, four in the Levantine Sea, two in the Black Sea, one in the Marmara Sea, one in Aegean Sea, two in the Ionian Sea, seven in the Adriatic Sea and two in the Tyrrhenian Sea (Figure 3).

**Table 1.** Location of the main coastal lagoons of the Mediterranean Sea Basin, indicating their name, numerical code (N) in Figure 3, country where they are located, sub-basin to which they belong (Sea), latitude (Lat), longitude (Long) in decimal degrees, perimeter in km, surface area in km$^2$, maximum depth (Depth) in m and width of the opening to the sea (Open) in m.

| Name | N | Country | Sea Basin | Lat. | Long. | Perim. | Area | Depth | Open |
|---|---|---|---|---|---|---|---|---|---|
| Berre | 1 | France | Lion gulf | 43.47 | 5.10 | 67.75 | 160.03 | 9 | 105 |
| Vaccares-Monro | 2 | France | Lion gulf | 43.54 | 4.58 | 56.91 | 134.62 | 2 | 59 |
| Or | 3 | France | Lion gulf | 43.58 | 4.03 | 29.05 | 36.43 | 4 | 114 |
| Perols-Vic | 4 | France | Lion gulf | 43.49 | 3.83 | 37.85 | 33.49 | 3 | 124 |
| Thau | 5 | France | Lion gulf | 43.40 | 3.61 | 48.67 | 70.91 | 10 | 65 |
| Bages-Sigean | 6 | France | Lion gulf | 43.08 | 3.01 | 43.39 | 57.86 | 2 | 111 |
| Leucate | 7 | France | Lion gulf | 42.85 | 3.00 | 41.64 | 56.76 | 4 | 20 |
| Saint Nazaire | 8 | France | Lion gulf | 42.84 | 2.99 | 14.60 | 10.04 | 5 | 91 |
| Clot | 9 | Spain | Balearic | 40.65 | 0.66 | 17.13 | 8.63 | 1 | 46 |
| Albufera | 10 | Spain | Balearic | 39.34 | −0.35 | 25.99 | 27.09 | 1 | 101 |
| La Mata-Torrevieja | 11 | Spain | Alboran | 38.01 | −0.72 | 35.48 | 30.51 | 3 | 13 |
| Mar Menor | 12 | Spain | Alboran | 37.73 | −0.79 | 52.67 | 136.90 | 8 | 186 |
| Marchica | 13 | Morocco | Alboran | 35.16 | −2.85 | 57.65 | 116.61 | 5 | 300 |
| Bizerte | 14 | Tunisia | Sicily | 37.19 | 9.86 | 52.02 | 128.82 | 8 | 232 |
| Ghar el Melh | 15 | Tunisia | Sicily | 37.15 | 10.18 | 25.24 | 36.39 | 2 | 74 |
| Tunis | 16 | Tunisia | Sicily | 36.82 | 10.25 | 48.99 | 68.54 | 1 | 291 |
| Boughrara | 17 | Tunisia | Gabès gulf | 33.60 | 10.80 | 138.14 | 536.13 | 16 | 2202 |
| El Bibane | 18 | Tunisia | Gabès gulf | 33.25 | 11.25 | 76.65 | 232.97 | 5 | 914 |
| Burullus | 19 | Egypt | Levantine | 30.89 | 31.48 | 124.63 | 512.98 | 2 | 80 |
| Manzala | 20 | Egypt | Levantine | 31.31 | 31.99 | 172.46 | 837.98 | 1 | 280 |
| Bardawil | 21 | Egypt | Levantine | 31.15 | 33.20 | 224.43 | 626.30 | 5 | 578 |
| Akyatan | 22 | Turkey | Levantine | 36.63 | 35.26 | 47.43 | 86.66 | 1 | 180 |
| Karine | 23 | Turkey | Aegean | 37.59 | 27.18 | 30.88 | 34.58 | 1 | 680 |
| Balik | 24 | Turkey | Black | 41.58 | 36.08 | 25.77 | 24.07 | 2 | 185 |
| Razim-Sinoie | 25 | Romania | Black | 44.80 | 29.00 | 223.72 | 1291.86 | 4 | 40 |
| Kucukcekmece | 26 | Turkey | Marmara | 41.01 | 28.75 | 24.63 | 15.22 | 20 | 25 |
| Amvrakikos | 27 | Greece | Ionian | 38.99 | 20.92 | 168.26 | 551.95 | 58 | 590 |
| Nartes | 28 | Albania | Adriatic | 40.54 | 19.43 | 33.51 | 55.40 | 1 | 60 |
| Karavasta | 29 | Albania | Adriatic | 40.93 | 19.49 | 50.79 | 82.48 | 1 | 100 |
| Marano-Grado | 30 | Italy | Adriatic | 45.74 | 13.20 | 80.11 | 160.96 | 6 | 1788 |
| Venice | 31 | Italy | Adriatic | 45.40 | 12.30 | 148.20 | 563.69 | 22 | 1730 |
| Comacchio | 32 | Italy | Adriatic | 44.60 | 12.18 | 51.79 | 147.53 | 2 | 48 |
| Lesina | 33 | Italy | Adriatic | 41.88 | 15.45 | 52.23 | 51.01 | 3 | 27 |
| Varano | 34 | Italy | Adriatic | 41.88 | 15.74 | 34.97 | 65.52 | 1 | 56 |
| Mare Piccolo | 35 | Italy | Ionian | 40.48 | 17.28 | 25.17 | 21.20 | 13 | 142 |
| Orbetello | 36 | Italy | Tyrrhenian | 42.45 | 11.21 | 42.64 | 33.47 | 1 | 70 |
| Cabras | 37 | Italy | Tyrrhenian | 39.94 | 8.48 | 33.75 | 23.66 | 2 | 204 |

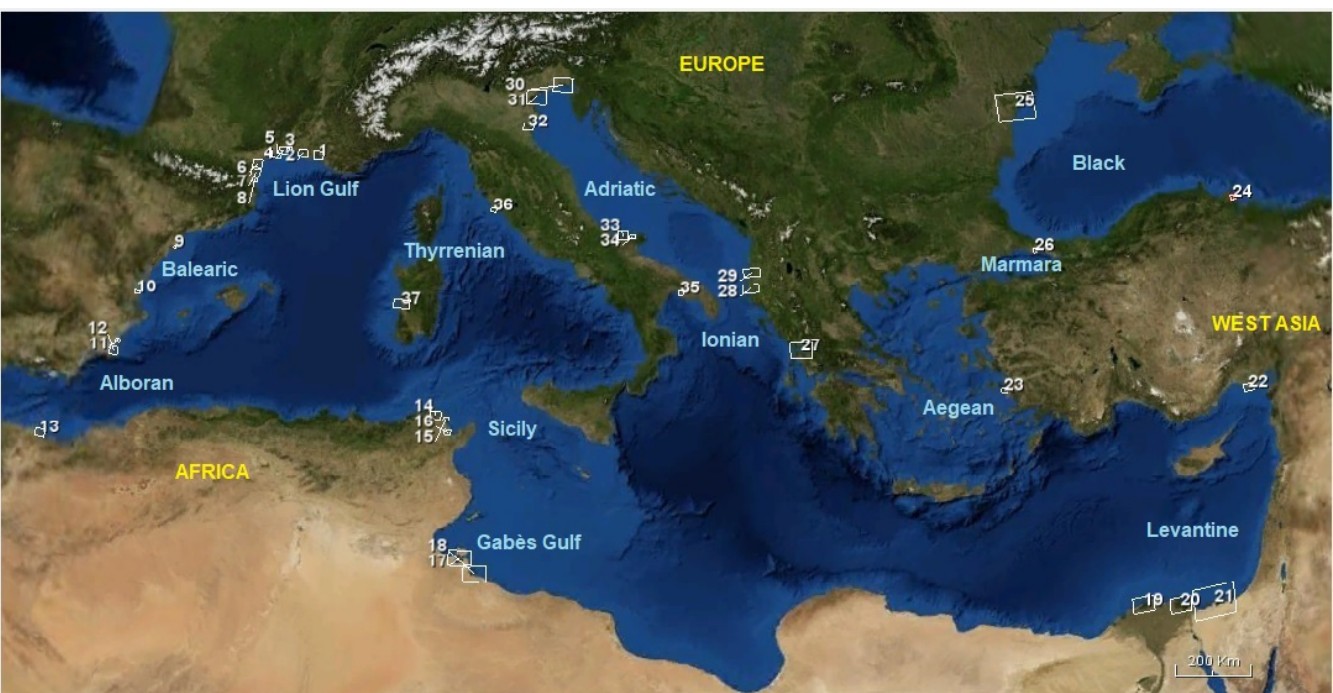

**Figure 3.** Location of Mediterranean Sea basins and coastal lagoons, indicating their numerical code according to Table 1. Base map source: GlobeView (https://globeview.nepia.com/, accessed on 15 January 2022).

In six cases, the lagoonal water body is geographically divided into two lagoons very close to each other, so that for the purposes of this review it is considered as one water body with two lagoons, each with its own name. This is the case, for example, of the Vaccarès and Monro lagoons (number 2) in France [24] or the Marano and Grado lagoons (number 30) in Italy [25].

Most lagoons are related to the presence of a deltaic river formation, such as the deltas of the Rhone, Ebro, Jucar, Segura, Moulouya, Nile, Danube and Po rivers. Some other lagoons have their origin in the movement of sands by marine currents, with the origin of materials from other places, such as the lagoons of Tunisia and those of Italy, whose formation is due to coastal morphology, without the presence of a river with deltaic formation. Figure 3 shows the location of the studied lagoons on the Mediterranean coast.

As for their characteristics, the coastal lagoons are mostly shallow, with maximum depths of up to ten meters, except those where the connection with the sea is open and are used for navigation, so they have up to 58 m as the deepest value [26] in the case of the Gulf of Amvrakikos (number 27). The maximum extension is presented in the Razim-Sinoie lagoon (number 25), located in the Danube delta [16], with an area of 1292 km$^2$. However, the longest is the Bardawil lagoon (number 21) located in the Nile delta [27], with 33.2 km on its longest axis. The image of each coastal lagoon obtained from a Landsat-8 scene during the summer of 2020 can be examined in Figure S1 (supplementary materials).

The bibliographic review has shown that the most studied are the Venice lagoon [28] (more than 1800 publications) and the Mar Menor (about 300) [29]. At a lower level, the lagoons of Thau, Leucate, Bizerte, Tunis, Marano-Grado and Orbetello already have between 300 and 100 publications (Figure 4a) [25,30–34].

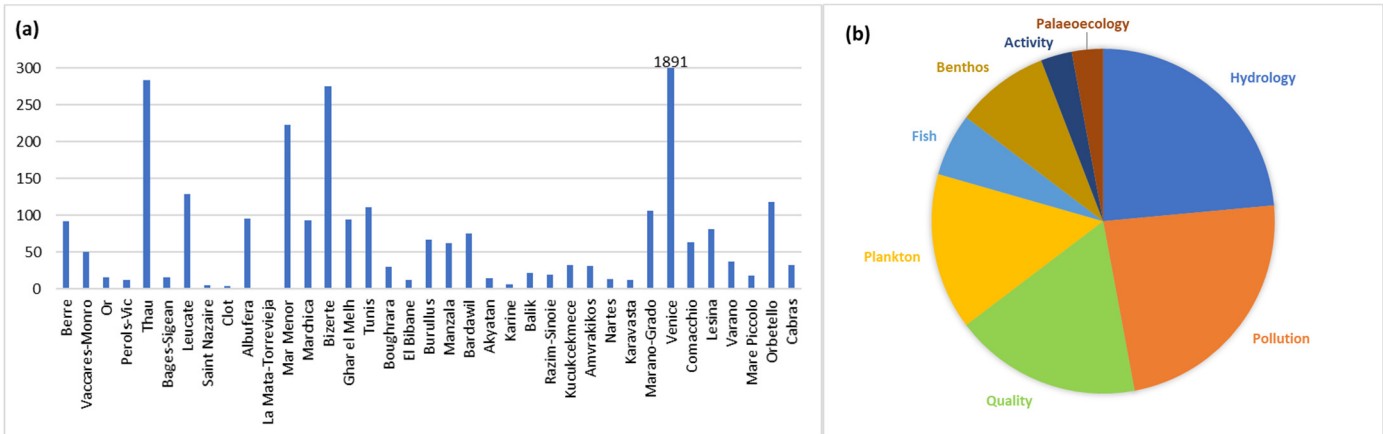

**Figure 4.** (**a**) Number of scientific publications indexed on each of the Mediterranean coastal lagoons. (**b**) Scope of the recent papers from 2016 to 2021 published in each lagoon.

Of the most recent published papers, the main topics are related to hydrology, pollution and water quality (Figure 4b), while the remaining third of the publications are related to plankton, fish, sediment and benthos, economic activities and paleoecology.

Regarding the trophic state, the results obtained (Table A1) indicate that the lagoons are generally in a poor trophic state (Figure 5). From the chlorophyll *a* content, applying the trophic state ranges defined by Carlson and Havens [35], eight lagoons are eutrophic (chlorophyll concentration between 8 and 25 µg/L) and 13 exceed this value, being classified as hypertrophic. Ten of them are mesotrophic and only six are in a good trophic state, with a chlorophyll *a* value of less than 2 µg/L: Berre (France), Mar Menor in Spain, Marchica (Morocco), Amvrakikos in Greece and Varano and Mare Piccolo in Italy. These are characterized by a good openness to the sea (except Varano). One might think that those lagoons with good communication with the sea are in a better condition than those with more limited communication, but it is clear that the pressures of the environment are more important [6,13].

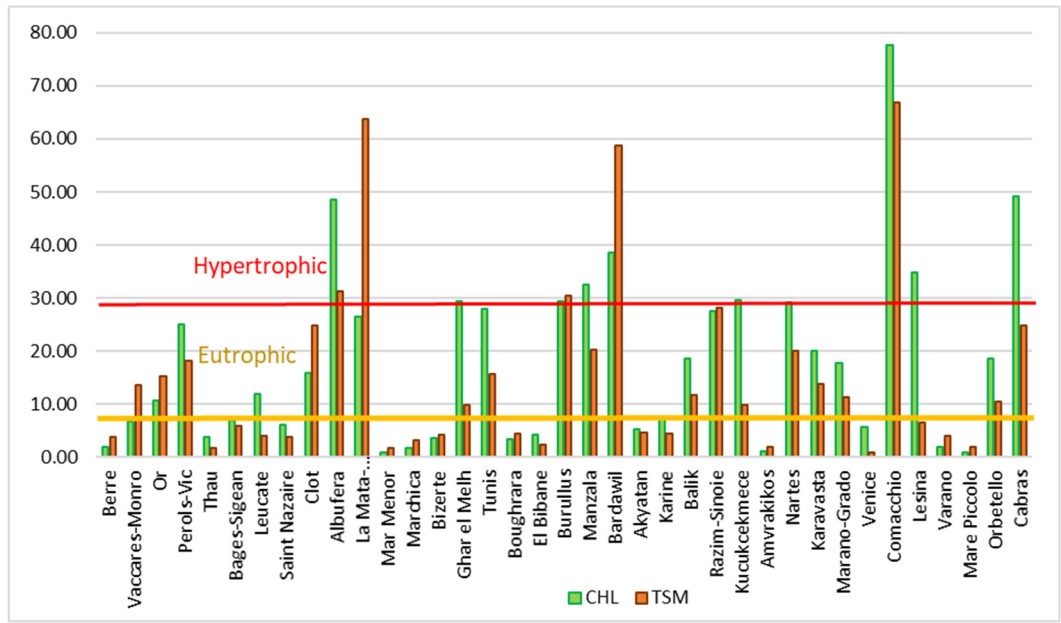

**Figure 5.** Trophic status of coastal lagoons assessed by chlorophyll *a* concentration in summer 2020. Chlorophyll *a* concentration (CHL) in mg/m$^3$ and TSM concentration in mg/L.

There is a significant statistical correlation between water transparency, measured as coefficient kd_z90max, and CHL and TSM concentration, in both cases $p < 0.001$, but the coefficient of determination is much higher for CHL than for TSM. High transparency values of water are correlated with low values of phytoplankton and suspended matter in the water (which can be biological or inert material). Figure 6 shows the distribution of the values of both variables.

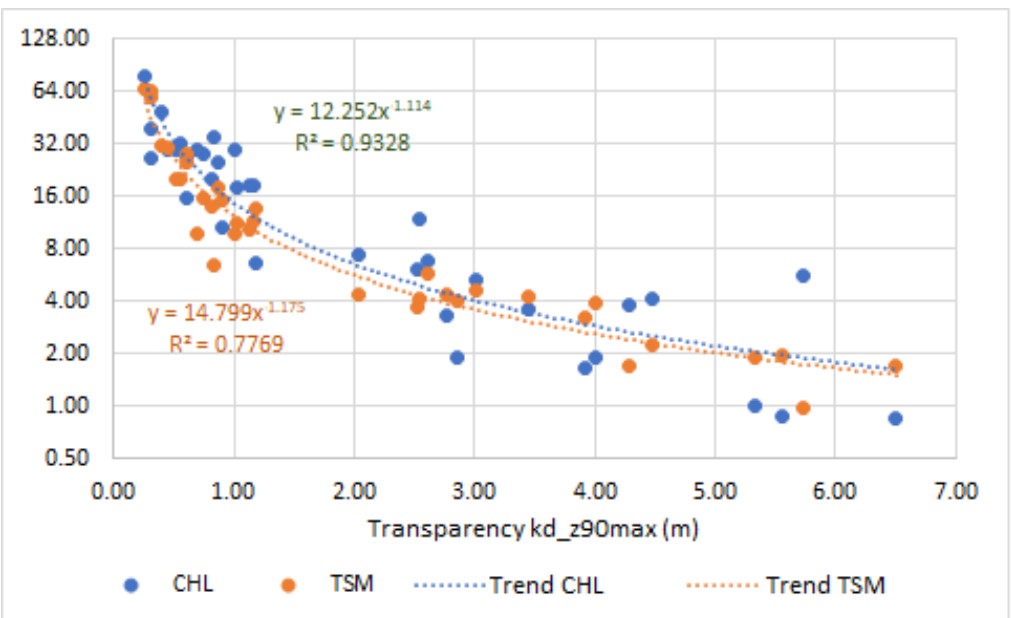

**Figure 6.** Representation of water transparency values (kd_z90max) versus chlorophyll *a* concentration in mg/m$^3$ (CHL) and suspended matter in mg/L (TSM). For each variable, the trend line, the adjust equation and its coefficient of determination are represented.

Ecologically, their importance lies in the fact that they are fluctuating environments, where aquatic vegetation of interest such as *Ruppietea maritimae*, *Potametea*, *Zosteretea* or *Charetea* associations develop [36]. The variation in salinity favors the presence of certain catadromous species of fish, so the use of these bodies of water for fishing by humans is historical. It is curious that the fishing systems used in the Mediterranean are similar in both Europe and Africa, performing set-ups with palisades and fixed nets that lead to the displacement of fish to the point where they are caught by the capture and extraction nets. From the aerial images it can be seen that in all locations the set-up is similar (Figure 7). It has been shown from research that the fishing system is of Phoenician origin, and that is why it has been widespread throughout the Mediterranean for more than 2000 years. However, for Steinberg [37], the system was already in use about 4500 years ago in Denmark (Figure 7d).

Beyond the environmental value of these treats, the cultural and scenic value of the coastal lagoons is recognized by the citizens of the surrounding area [38]. First of all, by the fishing exploitation of local communities since historical times [10,39], and also the scenery of the water from the shore, in conjunction with the visualization of the environment, makes it highly appreciated, not only for the richness in species, but also for the visualization of the sunrise or sunset. In many lagoons there are a large number of landscapes related to this spiritual appreciation (Figure 8). Table A2 in the annex presents a list of the existing environmental protection figures as well as the tourist values that are of cultural interest.

**Figure 7.** Current fishing set-up systems used in Mesopotamia in various coastal lagoons: (**a**) Redolín in Albufera de Valencia (Spain) March 2020. (**b**) Charfia in El Bibane (Tunisia). (**c**) Encañizada in Mar Menor (Spain) March 2017 (Archive of Murcia County Government). (**d**) Reconstruction of a fish hedge from Oleslyst, Denmark, about 4500 years old, sketch traced by J.S. [37].

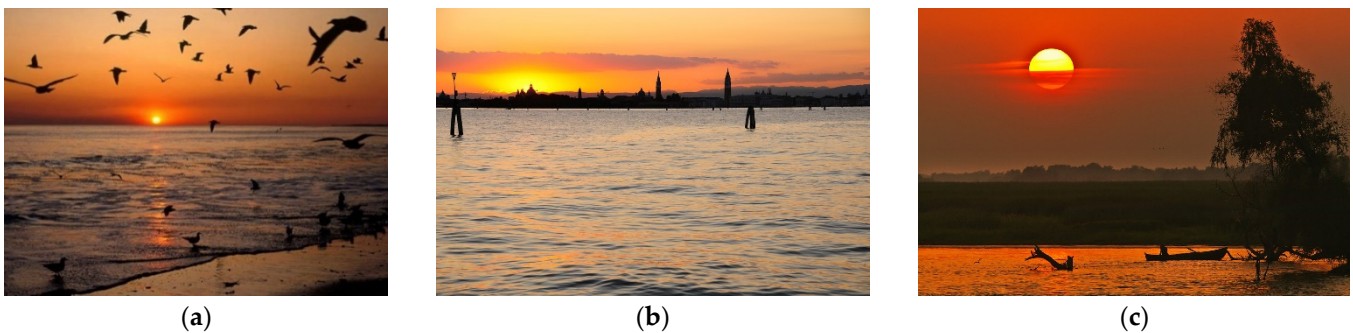

(**a**)            (**b**)            (**c**)

**Figure 8.** Sunset in some lagoons: (**a**) Bardawil, (**b**) Venice, (**c**) Razim-Sinoie. Source: Google Photos.

## 4. Discussion

The sites with the most publications are those where there is the existence of environmental problems of the site. Both lagoons (Venice and Mar Menor) coincide with two places with a very important tourist and agricultural impact in their basin. While in Venice, the uses are historical, having been cited for hundreds of years, and the environmental problems have existed for a hundred years, in Mar Menor the impacts have been occurring

for fifty years, but in recent years they have been directed towards the eutrophication of the lagoon, mainly due to the increase in eutrophication caused by high intensity agricultural practices [8,9].

The problem of eutrophication is the most important impact reported in coastal lagoons, especially in those with inland water inputs and less exchange with the sea. There are publications related to this ecological problem both in lagoons located in Europe [40,41] and in Africa [42,43]. In general, the problem of eutrophication is related to human activities and hydrological processes that transport nutrients from nearby areas [1].

Traditional fishing uses have been decreasing in coastal lagoons as a consequence of the loss of quality and connectivity with the sea [44]. Significant correlations have been found between increased trophic status and declining fisheries [45]. However, the connectivity with the sea is the most important factor. On the one hand, if the opening is excessive, the environment becomes purely marine, rather than lagoonal, and this produces a loss of biodiversity [46]. However, when connectivity is lost or hindered by regulation of exchange, the loss is even greater; it is necessary to maintain the mosaic of lagoon ecosystems in order to conserve traditional fisheries [47].

Studies on the plankton are also frequent in the naturalistic descriptive of the lagoons, highlighting the works on taxonomy and the distribution of phytoplankton species [48,49]; secondly, there are studies related to zooplankton [50–53], the trophic structure of the community [54] as well as other more detailed studies on some endangered species [55]. Much more novel are those studies related to the use of eDNA as risk detection in lagoons [56].

The sediment of these lagoons, although quite recent, has served as an object of study to understand the changes that have taken place in the last thousands of years, as well as a record of the contamination received from the basin. Among these studies are those of the Albufera of Valencia [57], in which salinity changes in the lagoon are observed, the presence of contaminating metals such as in Bages [58] and Ghar El Melh [59], and the relationship between the conditions of the benthos and hypoxia in the water of the Berre lagoon [60].

The problem of navigation is another important impact, especially in the shallower lagoons. It has been studied how navigation affects sediment resuspension [61] in such a way that it is as important as that produced by breezes. On the other hand, the effect that navigation waves have also affects shoreline erosion in places with high traffic such as Venice [62]. Navigation inside the lagoons, not only for fishing, has also been a historical activity when there is an opening to the sea; this makes inland ports where the stay of ships is ensured, and therefore also affects water quality; in the case of Marano, for example, it has required dredging the bottom to provide navigation channels [63]. In other lagoons it has required opening the channel of communication with the sea to maintain a significant and deep opening, as in lagoons in southern France, in Marchica [64] and Bizerte [65]. This allows the renewal to be higher and with it a change in water quality in some areas of the lagoon [66].

A further problem, resulting from both globalization and climate change, is invasive species. Alien species can have different (negative) impacts on their new environment: (i) they can actively compete with autochthonous species (and in some cases eradicate them); (ii) they can be a vector for viruses and germs and favor epidemics, (iii) they can become invasive and create environmental problems [10].

Climate change has as a consequence on sea level rise and with it the disappearance of the sandbars due to shoreline recession [67]. These studies are the most recent on coastal lagoons, and in our study area have dealt especially with Venice [68], but are of concern in general in all coastal lagoons, especially those with urban settlements on their shores [69]. The results show that the level of the lagoons will also rise, as well as there being an increase in temperatures and salinity (especially in those that are less open to the sea) and, therefore, there will be a change in species [70] and a disappearance of lagoons that lose the coastline. The changes that lagoons undergo due to extreme events, such as floods [71,72] or pollution episodes, are also important [73], and in many cases lead to the implementation

of restoration measures [74] in order to preserve, in the best possible condition, these places of ephemeral life in geological time.

At present, coastal lagoons are places of great environmental interest for the protection of biodiversity, especially related to waterfowl and traditional fishing. This activity is in the minority and of little economic importance, but it belongs to the cultural heritage of civilization and must be maintained. The Ramsar Convention has designated practically all the lagoons as Protected Areas (see Table A2). It would be desirable to include also those that are not yet, such as the Nartes lagoon [75].

The progressive disappearance of the coastal lagoons is important not only for the environmental loss but also for the natural heritage, together with the economic and social losses in the region. In some cases, such as the Cabras lagoon, the environment has been colonized since the Paleolithic, and all Mediterranean civilizations have left their mark [76]. The poor quality of ecosystems today is closely related to eutrophication by human activity.

In some lagoons, situations have been reached that may be irreversible in the ecological processes due to existing uses. Aquaculture in these bodies of water can produce impacts that are difficult to recover, while the exchange of lagoons with the sea continues to be the most critical factor for quality [77]. In general, the studies propose in these sites that improved communication between the lagoon and the sea would be the most important improvement process; while in other water bodies, increased intercommunication with the sea is related to increased activities that worsen the lagoon [6,12,24].

## 5. Conclusions

The review of the existing coastal lagoons in the Mediterranean Sea basin whose surface area is larger than 10 km$^2$ provides a list of thirty-seven water bodies, distributed in Europe, Africa and Asia Minor. The three largest are Razim-Sinoie, located in the Danube delta in the Black Sea sub-basin, with a surface area of more than 1200 km$^2$. It is followed by two lagoons located in the Nile delta in the Levantine Sea sub-basin, Manzala with an area of about 800 km$^2$ and Bardawil, which exceeds 600 km$^2$. As might be expected, it is the deltaic formation that is responsible for the formation and conservation of these coastal lagoon areas, which are linked to the presence of a river with important contributions to the sea, as in the cases mentioned above and the lagoons located in the deltas of the Moulouya, Segura, Ebro, Rhone and Po rivers.

The trophic state of most of these lagoons is eutrophic or worse, with only three good quality lagoons, the Mar Menor (Spain), Amvrakikos (Greece) and Mare Piccolo (Italy). The eutrophication of the waters due to anthropogenic inputs is the main cause of their poor quality.

Communication with the sea and depth are the morphological variables that most affect the quality of the environment and its traditional and current uses. On the one hand, communication favors exchange and renewal, but it makes it lose its differential environment with respect to the Mediterranean. The depth increases the volume of the water mass and gives it stability, but favors navigation and related uses such as the establishment of ports in the interior, which also do not benefit the quality of the water.

These threatened habitats need real protection, leading to a resource management that favors the entry of quality water, and that the exchange with seawater is consistent with the inflows of freshwater to preserve its uniqueness and biodiversity, as well as the contributions of inland waters that are of good quality so that their trophic status does not worsen.

**Supplementary Materials:** The following supporting information can be downloaded at: https://www.mdpi.com/article/10.3390/jmse10030347/s1, Figure S1: False color image of each Mediterranean lagoon studied in summer 2020. Base bands RGB are 4, 3 and 2 of satellite Landsat-8 OLI mission. Product downloaded from Google Earth Engine Explorer.

**Author Contributions:** J.S. and R.P. conceived and designed the experiments; X.S.-P. performed the investigation; J.S. and R.P. analyzed the data; J.S. and R.P. wrote the paper; J.S. and X.S.-P. review and editing. All authors have read and agreed to the published version of the manuscript.

**Funding:** This research received no external funding.

**Institutional Review Board Statement:** Not applicable.

**Data Availability Statement:** Satellite images are available from the web site Google Earth Engine (https://earthengine.google.com/, accessed on 15 January 2022) and Copernicus Data Hub (https://cophub.copernicus.eu/dhus/, accessed on 20 November 2021).

**Conflicts of Interest:** The authors declare no conflict of interest.

## Appendix A

**Table A1.** Variables obtained by studying Sentinel-2 imagery in summer 2020. CHL-a, chlorophyll *a* (mg/m$^3$); TSM, total suspended matter (mg/L); Kd_z90max, water transparency measured as depth at which 10% of incident light reaches the surface (m). Highlighted in green, lagoons in better ecological status; in orange, lagoons in worse ecological status.

| Name | Number | Country | Sea | CHL-a | TSM | Kd_z90max |
|---|---|---|---|---|---|---|
| Berre | 1 | France | Lion gulf | 1.91 | 3.88 | 4.01 |
| Vaccares-Monro | 2 | France | Lion gulf | 6.65 | 13.60 | 1.19 |
| Or | 3 | France | Lion gulf | 10.71 | 15.33 | 0.90 |
| Perols-Vic | 4 | France | Lion gulf | 25.02 | 18.12 | 0.86 |
| Thau | 5 | France | Lion gulf | 3.82 | 1.71 | 4.28 |
| Bages-Sigean | 6 | France | Lion gulf | 6.83 | 5.79 | 2.61 |
| Leucate | 7 | France | Lion gulf | 11.93 | 4.12 | 2.54 |
| Saint Nazaire | 8 | France | Lion gulf | 6.16 | 3.72 | 2.53 |
| Clot | 9 | Spain | Balearic | 15.86 | 24.83 | 0.61 |
| Albufera | 10 | Spain | Balearic | 48.64 | 31.31 | 0.39 |
| La Mata-Torrevieja | 11 | Spain | Alboran | 26.52 | 63.83 | 0.31 |
| Mar Menor | 12 | Spain | Alboran | 0.85 | 1.70 | 6.50 |
| Marchica | 13 | Morocco | Alboran | 1.65 | 3.24 | 3.92 |
| Bizerte | 14 | Tunisia | Sicily | 3.60 | 4.23 | 3.44 |
| Ghar el Mehl | 15 | Tunisia | Sicily | 29.40 | 9.80 | 1.00 |
| Tunis | 16 | Tunisia | Sicily | 27.88 | 15.61 | 0.74 |
| Boughrara | 17 | Tunisia | Gabès gulf | 3.37 | 4.36 | 2.76 |
| El Bibane | 18 | Tunisia | Gabès gulf | 4.17 | 2.28 | 4.48 |
| Burullus | 19 | Egypt | Levantine | 29.43 | 30.54 | 0.45 |
| Manzala | 20 | Egypt | Levantine | 32.45 | 20.21 | 0.55 |
| Bardawil | 21 | Egypt | Levantine | 38.58 | 58.80 | 0.31 |
| Akyatan | 22 | Turkey | Levantine | 5.32 | 4.66 | 3.01 |
| Karine | 23 | Turkey | Aegean | 7.43 | 4.34 | 2.04 |
| Balik | 24 | Turkey | Black | 18.49 | 11.62 | 1.16 |
| Razim-Sinoie | 25 | Romania | Black | 27.62 | 28.19 | 0.60 |
| Kucukcekmece | 26 | Turkey | Marmara | 29.53 | 9.89 | 0.70 |
| Amvrakikos | 27 | Greece | Ionian | 1.01 | 1.92 | 5.33 |
| Nartes | 28 | Albania | Adriatic | 29.26 | 20.08 | 0.52 |
| Karavasta | 29 | Albania | Adriatic | 19.99 | 13.87 | 0.81 |
| Marano-Grado | 30 | Italy | Adriatic | 17.81 | 11.31 | 1.03 |
| Venice | 31 | Italy | Adriatic | 5.70 | 0.98 | 5.73 |
| Comacchio | 32 | Italy | Adriatic | 77.69 | 66.84 | 0.26 |
| Lesina | 33 | Italy | Adriatic | 34.75 | 6.53 | 0.84 |
| Varano | 34 | Italy | Adriatic | 1.93 | 4.02 | 2.86 |
| Mare Piccolo | 35 | Italy | Ionian | 0.87 | 1.98 | 5.55 |
| Orbetello | 36 | Italy | Tyrrhenian | 18.61 | 10.48 | 1.13 |
| Cabras | 37 | Italy | Tyrrhenian | 49.22 | 24.80 | 0.41 |

**Table A2.** Aspects of interest in the studied lagoons related to biodiversity, tourist interest, gastronomy and protection figures.

| Country | Name | Sea | Biodiversity | Tourism Interest | Gastronomy | Protection |
|---|---|---|---|---|---|---|
| France | Vaccares-Monro | Lion gulf | There are 65% unique species in the area (flora and fauna) | City of Marseille, medieval towns and Avignon | Vineyards, wines of Luberon, olive oil, lavender oil, and cheeses | Ramsar site. Natura 2000. Department of Bouches-du-Rhone. Provence natural parks. |
| | Le Berre | | | | | |
| | Thau | | | | | |
| | Pérols-Vic | | Montpellier was chosen as the capital of French biodiversity | City of Montpellier, the *folies* (ancient palaces), the region of Languedoc-Rousillion | Wines, seafood and bull meat | Ramsar site. Natura 2000. Green Net (2007) which cares about studies of environmental impact. |
| | L'Or | | | | | |
| | Leucate | | In this county, there is one of the largest natural reserve of France. | Beaches, lagoons, vineyards | Seafood and steakhouses | Ramsar site. Natura 2000. |
| | Bages-Sigean | | | | | |
| | Saint Nazaire | | | | | |
| Spain | Clot | Balearic | Habitat of 280 species of birds | Cities of Tarragona and Valencia, medieval towns, beaches | Seafood, paella, duck rice, "l'espardenyà" | Ramsar site. Natura 2000. Natural parks. |
| | Albufera | | | | | |
| | La Mata-Torrevieja | Alboran | Rich zone of biodiversity of marine and terrestrial species. Some are endemic and of great biological interest. | Cities of Alacant and Murcia, some popular towns, beaches | Michelin restaurants, seafood, mojama | Ramsar site. Natura 2000. Natural parks. |
| | Mar Menor | | | | | |
| Morocco | Marchica | Alboran | Flamingo bay, birds shelter | Resorts, golf course, cities of Nador and Melilla | Tajine, couscous, Khubz bread, hummus | Ramsar site. *Institut National de Recherche Halieutique-Centre de Nador* (University of Mohammed V) |

**Table A2.** *Cont.*

| Country | Name | Sea | Biodiversity | Tourism Interest | Gastronomy | Protection |
|---|---|---|---|---|---|---|
| Tunisia | Bizerte | Sicily | Ichkeul Natural Park (nature reserved named World Heritage by UNESO). | Qsiba, Ichkeul National Park, Utica, ottoman fortresses | Seafood, chakchouka, couscous, jelbana, kefteji | Ramsar site. National Tourism Organization |
| | Ghar el Mehl | | | | | |
| | Tunis | | | | | |
| | Boughrara | Gabès gulf | | | | |
| | El Bibane | | | | | |
| | Manzala | | | | | |
| | Bardawil | | | | | |
| Egypt | Burullus | Levantine | 135 endemic plants, dunes, natural reserves | Cities of El Cairo and Alexandria | Ful Medames, Koshari, Baba Ganoush, Fatteh, Kofta, falafel | Ramsar site. |
| | Manzala | | | | | |
| | Bardawil | | | | | |
| Turkey | Akyatan | Levantine | Important Bird Area | Adana, Mersin, ottoman castles | Ayran, Raki, Kahve, baklava, Turkish delicacies | Ramsar site. Natural Park |
| | Karine | Aegean | Aquatic birds | Buyuk delta | | |
| | Balik | Black | Birds, velvet duck | Estambul, Ankara | | |
| | Kucukcekmece | Ionian | | | | |
| Romania | Razim-Sinoie | Black | Birds and fishes | Museum of Ebisala, Istria, Aerganum fortresses, Sarichioi and Jurilovca towns | Ciorba de burta, Zacusca, Mici, sarmale | Ramsar site. World heritage site. UNESCO Biosphere Reserve |
| Greece | Amvrakikos | Ionian | Marshes, endemic flora and bids fauna | Arta, byzantine fortresses, island of Koronisia | Moussaka, tzatziki, gyros, fava | Ramsar site. Natura 2000. |
| Albania | Karavasta | Adriatic | Birds protection | Byzantine art | Fish and eels | Ramsar site. National Park. |
| | Nartes | | | | | |

**Table A2.** *Cont.*

| Country | Name | Sea | Biodiversity | Tourism Interest | Gastronomy | Protection |
|---|---|---|---|---|---|---|
| Italy | Marano-Grado | Adriatic | High biodiversity and there are a lot of marine regions protected. Endemic Flora. Fauna there are swans, dophins, friar seal or flamingos | Venice, Murano, Bolonia, Ferrera, Verona. Cultural heritage. | Pasta, pizza, wine, tiramisu, risotto, carpaccio, calzone | Natura 2000. Ramsar site.Protected by the statal government and UNESCO |
| | Venice | | | | | |
| | Comacchio | | | | | |
| | Lesina | | | | | |
| | Varano | | | | | |
| | Mare Piccolo | Ionian | | Tarento, churches | | |
| | Orbetello | Tyrrhenian | | Toscana, Porto Ercole and Porto Santo Stefano | | |
| | Cabras | Sardinia | Historic place | | Sardinia island | |

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
