# Peer review of "Mediterranean Coastal Lagoons Review: Sites to Visit before Disappearance"

_jmse, doi:10.3390/jmse10030347_

Round 1

Reviewer 1 Report

Interesting focus from biodiversity and evolutionary ecology.

About lagoons cited, a table should be included (Lagoons on de X axis and on Y axis options to visited (landscape, biodiversity, gastronomy, tourism, etc) to reinforce the scattered comments for greater utility in field of regional development and evironmental problems.

Include in the analysis more detail about how the temporaly gradual loos of the lagoons will affect these areas in the economically, socially and evironmentaly

Author Response

Interesting focus from biodiversity and evolutionary ecology.

About lagoons cited, a table should be included (Lagoons on de X axis and on Y axis options to visited (landscape, biodiversity, gastronomy, tourism, etc) to reinforce the scattered comments for greater utility in field of regional development and environmental problems.

Thank you very much for your valuable comments. We make a table with the interesting aspects and grouped by geographical area. The main table is located in the annex as table A2, and the main topics are discussed in the end of the section, before Conclusions.

Include in the analysis more detail about how the temporally gradual loos of the lagoons will affect these areas in the economically, socially and environmentally

This interesting point of view is also discussed after the paragraph above described, before Conclusions (lines 292-304)

Reviewer 2 Report

It is a potentially valuable review of Mediterranean coastal lagoons. However, I indentify two major drawbacks that the Authors should address before I can consider this manuscript further: (i) the benthic compartment is largely overlooked both in terms of benthic assemblages and man-made, eutrophication-related pollution (e.g. sediment organic enrichment); (ii) for a more correct and comprehensive geographical coverage and in line with the focus on Mediterranean lagoons “larger than 10 km2”, the review should include at least the largest Sardinian lagoon, Cabras lagoon, with a surface of 22 km2 (e.g. see Basset et al. 2006 Aquatic Conserv: Mar. Freshw. Ecosyst. 16: 441–455). For this lagoon, many publications on both the benthic and pelagic compartments are available which provide an extensive representation of the ecology and functioning of nanotidal lagoons greater than 10 km2 located in the western Mediterranean Sea. Indeed, several other Mediterranean lagoons greater than 10 km2 can be found, such as Logarou in Greece, Karavasta and Narta in Albania, but ecological data may be more limited here. For these reasons, statements given at lines 91-92 (ref. item ii) and lines 159-161 (ref. item i), as an example, are not correct and misleading. Thus, I encourage the Authors to consider the above points, to implement their dataset and to reassess relevant analysis and results accordingly.

I remain available for further review once those issues are accomplished.

Author Response

It is a potentially valuable review of Mediterranean coastal lagoons. However, I identify two major drawbacks that the Authors should address before I can consider this manuscript further: (i) the benthic compartment is largely overlooked both in terms of benthic assemblages and man-made, eutrophication-related pollution (e.g. sediment organic enrichment); (ii) for a more correct and comprehensive geographical coverage and in line with the focus on Mediterranean lagoons “larger than 10 km2”, the review should include at least the largest Sardinian lagoon, Cabras lagoon, with a surface of 22 km2 (e.g. see Basset et al. 2006 Aquatic Conserv: Mar. Freshw. Ecosyst. 16: 441–455). For this lagoon, many publications on both the benthic and pelagic compartments are available which provide an extensive representation of the ecology and functioning of nanotidal lagoons greater than 10 km2 located in the western Mediterranean Sea. Indeed, several other Mediterranean lagoons greater than 10 km2 can be found, such as Logarou in Greece, Karavasta and Narta in Albania, but ecological data may be more limited here. For these reasons, statements given at lines 91-92 (ref. item ii) and lines 159-161 (ref. item i), as an example, are not correct and misleading. Thus, I encourage the Authors to consider the above points, to implement their dataset and to reassess relevant analysis and results accordingly.

Thank you very much for your contributions. Logarou lagoon are included for us in the complex wetland of Amvrakikos, that includes the gulf of Arta, and the lagoons that are separated by small sand barriers from the main body water.

We have included in the list and in the study the other indicated: Nartes lagoon, Karavasta lagoon, Cabras lagoon.

About the benthic compartment, we have included in a new search this aspect. We include some comments about in lines 252-257.

Figures and tables are modified to include the new lagoons.

Reviewer 3 Report

Figure 2, It is difficult to understand the stratification of saltwater and freshwater because two waters are drawn with lines. It is better to in the water area with color.

Line 110, Please introduce methodology for estimating Chl-a, TSM and kd_z90max in detail, or introduce the paper you cited. Did you collect both in-situ data of Chl-a, TSM and kd and satellites data, then investigate correlation between them, and calculate Chl-a, TSM and kd distribution by using correlation?

Figure 6, Please show title and unit of X, Y axis.

Line 185-190, The significance of this sentence to the whole is unclear. What is the meaning of “The contribution of Chl to water transparency is greater than that of TSM” in the Mediterranean Sea?

Author Response

Figure 2, It is difficult to understand the stratification of saltwater and freshwater because two waters are drawn with lines. It is better to in the water area with color.

We have modified the figure and added a vertical profile of salinity, and minor changes in lines to be more comprehensive.

Line 110, Please introduce methodology for estimating Chl-a, TSM and kd_z90max in detail, or introduce the paper you cited. Did you collect both in-situ data of Chl-a, TSM and kd and satellites data, then investigate correlation between them, and calculate Chl-a, TSM and kd distribution by using correlation?

The methodology is reworded and better described in section 2 (paragraph 111-122): “The atmospheric correction has been processed using the Case 2 Regional Coastal Color tool together with the Case 2 Extreme neural networks for complex waters, after resampling the images so that their bands have the same spatial resolution of 10 meters. Once the resampling and correction procedure was done, the automatic SNAP products are obtained, which are the concentration of Chlorophyll a (CHL), the concentration of total suspended matter (TSM) and the transparency of the water measured as the coefficient kd_z90max, the depth at which 10% of the incident radiation reaches the surface. The values obtained with this procedure are validated by numerous published papers.”

Figure 6, Please show title and unit of X, Y axis.

We add the title of X axis in the figure. Due to the Y axis is the same scale for CHL and TSM but the units are different, the unis are detailed in the figure caption.

Line 185-190, The significance of this sentence to the whole is unclear. What is the meaning of “The contribution of Chl to water transparency is greater than that of TSM” in the Mediterranean Sea?

We have explained better: High transparency values of water are correlated with low values of phytoplankton and suspended matter in the water (which can be biological or inert material). Figure 6 shows the distribution of the values of both variables (lines 191-194).

Round 2

Reviewer 2 Report

I appreciate the effort made by the Authors to address my criticisms. In the revised version, the Authors provide a more extensive and balanced overview of Mediterranean lagoons both quantitatively (e.g. Figures 4, 5 and 6 and Table A1) and qualitatively (e.g. Table A2). The discussion and the overall MS have been improved.

Finally, a few minor issues to be considered by the Authors:

  • Abstract: “34 coastal lagoons” change to “37 coastal lagoons”.
  • Lines 108-110. “Using the Scopus bibliographic review…”: the time-frame (years) of this search should be indicated.
  • Line 161: “…the more recent papers …”, please indicate the time-frame (years).
  • Line 307: “a list of thirty-four water bodies” change to “a list of thirty-sever water bodies”.

I congratulate the Authors for the interesting work done.

Author Response

Thank you very much for your detailed observations. Your work as reviewer contributes to the best of the manuscript. This minor observations indicates that is  a good review.

We have changed the 34 to 37 (we add in first review 3 more lagoons and in some places was the old number).

We add the time framework of the search (1970 to present) and the recent papers from 2016 to present.